# Health-Related Quality of Life in Migraine: EQ-5D-5L-Based Study in Routine Clinical Practice

**DOI:** 10.3390/jcm11236925

**Published:** 2022-11-24

**Authors:** Izabela Domitrz, Dominik Golicki

**Affiliations:** 1Department of Neurology, Faculty of Medical Sciences, Medical University of Warsaw, 80 Cegłowska St., 01-809 Warsaw, Poland; 2Department of Experimental and Clinical Pharmacology, Medical University of Warsaw, Żwirki i Wigury 61, 02-091 Warsaw, Poland

**Keywords:** episodic migraine, chronic migraine, quality of life, EQ-5D-5L

## Abstract

Background: Migraine leads to moderate to severe disabilities and disrupts family life, interpersonal relationships, and professional life, and is the second leading cause of disability worldwide. Many people with migraine suffer prolonged headaches and frequent migraine attacks, transition to having chronic migraine, and have the highest number of disability-adjusted life-years. The aim of this study is to measure the quality of life in migraineurs based on the EQ-5D-5L questionnaire. Methods: We assessed 100 consecutive patients diagnosed with migraine: 70 with episodic migraine and 30 with chronic migraine. Migraineurs were asked to complete the EQ-5D-5L. The control group (n = 100), matched for sex and age group, was created based on the results of the population norms study for the EQ-5D-5L in the general population of Poland. Results: Patients with migraine had worse HRQoL than the matched general population control group for all three primary endpoints of the EQ-5D-5L questionnaire: dimensions, EQ-5D-5L Index and EQ VAS. Conclusions: Migraine is a disease that disrupts daily function, and as a lifelong disease, plays a role in every aspect of it. Proving a negative impact on many aspects helps to make decisions about treatment, especially in the context of the design and reimbursement of drugs.

## 1. Introduction

Neurological disorders are an important cause of disability and death worldwide [1]. Migraine, as a neurological disease, is a common primary headache that reduces the quality of life, increases the economic burden, and weakens production capacity [2,3]. Migraine is undoubtedly one of the best-defined pain diseases and is associated with the highest number of disability-adjusted life-years (DALYs) [4]. It often leads to moderate to severe disabilities and disrupts family life, interpersonal relationships, and professional life [5], and is the second leading cause of disability worldwide. It is important to understand the migraine experience by endeavoring to understand the daily life, real needs and personal resources of migraineurs, their caregivers and clinicians within the evolution of the care pathway, as shown in the few studies so far [6]. There is a paucity of robust data on migraine-related work factors associated with productivity [7]. Many people with migraine suffer prolonged headaches and frequent migraine attacks, and transition to chronic migraine (CM) despite prophylactic treatments [8].

Almost 2% of the general population suffers from CM, the most severe and disturbing type of migraine, which develops from an episodic one [9]. The 12-year study in Copenhagen showed an annual migraine incidence of 8.1/1000 with a female-to-male ratio of 6.2:1 [10]. The cumulative lifetime incidence is 43% for women and 18% for men. Peak migraine incidence is achieved at 20–24 years of age for women and 15–19 years of age for men.

The overall prevalence of chronic migraine ranges from 1.4% to 2.2% of the general population, which reflects 8% of all individuals suffering from migraine [11]. Chronic migraine is 4.57 times more prevalent in women than in men. In women, the prevalence of chronic migraine peaks at 18–29 years of age and again at 40–49 years of age [12]. Chronic migraine mostly develops from episodic migraine with a conversion rate of 2.5% to 3% per year [13]. Another critical issue in the context of chronic migraine is the dangerous complication of medication overuse [14]. Drug addiction could either lead to an increase in the risk or the manifestation of medication overuse. Drug addiction itself is dangerous for patients, and it is known to be associated with drug-induced headache or medication-overuse headache, which again impair daily functioning and promote drug abuse. An opioid prescription must be completely avoided considering its high potential for abuse and high economic and societal costs [14].

The social cost of migraine is high for several reasons. These include inefficiency at work, reduced functionality, medication use, and outpatient, inpatient and emergency medical visits. Migraine, like any disease entity, comes with costs, and such an analysis has been made recently in Poland [15]. The conclusion is, therefore, that the primary cost implication of migraine is the cost of significantly reduced productivity. Most people with migraine attacks come to work or develop migraine attacks at work and then carry out their duties and activities with significantly limited productivity. This limitation can be as high as 50% of the performance loss or more. It is necessary to conduct education in society and to develop optimal standards of care for patients suffering from migraine in order to minimize its health, social and economic effects. In this regard, because migraine is a frequent, disabling, chronic disorder with a significant impact on patient well-being, many questionnaires and tests have been introduced to assess the burden of this disease. Many scales and questionnaires evaluate the quality of life of migraine patients. Patient-reported outcome measures (PROMs) fall into two classes: generic measures and disease-specific measures. Generic PROMs, such as the Short-Form 36-item health profile (SF-36) [16], its shorter derivative the Short-Form 12-item health profile (SF-12) [17], the Sickness Impact Profile, the Nottingham Health Profile [18] and, more recently, the EuroQoL five-dimension questionnaire (EQ-5D) [19] have been developed. Disease-specific measures in migraine, such as the Migraine-Specific Quality of Life Questionnaire [20] and the 6-item Headache Impact Test (HIT-6) [21], are well established and validated. Some authors have introduced new survey scales. Lipton et al. [22], this year, evaluated the content validity and psychometric properties of the Activity Impairment in Migraine Diary (AIM-D). The authors stated that measuring treatment effects on migraine impairment requires a psychometrically sound patient-reported outcome measure consistent with U.S. Food and Drug Administration guidance [22]. The multicohort longitudinal survey called the Observational Survey of the Epidemiology, Treatment and Care of Migraine (OVERCOME; USA) study [23] assesses symptomatology, consulting, diagnosis, treatment, and impact of migraine in the United States. Regularly updating population-based views of migraine provides a method for assessing the quality of ongoing migraine care and identifying unmet needs [23]. Ferreira et al. [24] studied 92 patients, mainly female, with a mean age of 44 years and, on average, 9.7 headache days in the previous month, and pain averaging 7.5/10. Almost 70% were on a migraine prophylactic treatment, and more than 40% had a severe disability with anxiety and depression. Content validity showed that mMIDAS-P (modified Migraine Disability Assessment in the Portuguese population) is simple and clinically useful. It was not shown to be determined by the patient’s sociodemographic characteristics, and it was correlated with the depression scale and EQ-5D-5L. Test–retest demonstrated high reproductive reliability and good internal consistency. The authors concluded that mMIDAS-P is valid and reliable and strongly recommend it for clinical and research use [24]. 

In the study of Lucas et al. [25], 249 patients completed the EQ-5D-5L, the Work Productivity and Activity Impairment Questionnaire (WPAI), the Hospital Anxiety and Depression scale (HAD) and the 6-item Headache Impact Test (HIT-6). Low EQ-5D-5L utility scores were associated with frequent (≥15 headache days/month) or disabling (HIT-6 score ≥60) headaches. This study [25] involved patients with severe migraine, defined by the authors as reporting headaches on ≥8 days/month and having failed ≥2 prophylactic treatments. 

In fact, the EQ-5D is, apart from the Short Form-36 (SF-36), one of the most popular generic instruments for the measurement of health-related quality of life (HRQoL) [26,27,28]. The questionnaire is available in two versions: the original, three-level (EQ-5D-3L) and the more recent five-level form (EQ-5D-5L). The EQ-5D-5L vs. the EQ-5D-3L possesses some psychometric advantages, including a lower ceiling effect and higher sensitivity. A Polish validation of the EQ-5D-5L has recently been published [29,30,31]. The use of the EQ-5D in Poland is supported by the availability of many country-specific tools [32]. 

Our study aimed to assess the influence of migraine on the HRQoL of patients with EM or CM based on the EQ-5D-5L generic questionnaire. The HRQoL of migraine individuals was compared with a matched control coming from general population. 

## 2. Patients and Methods

### 2.1. Study Groups 

This cross-sectional study was conducted between September 2019 and December 2020 among migraine patients of the Headache Outpatient Clinic in Warsaw, Poland. Migraine was diagnosed according to the International Classification of Headache Disorders (ICHD)-3 [33].

#### 2.1.1. Study Group with Migraine

One hundred consecutive patients diagnosed with migraine—89 women (89%) and 11 men (11%)—were included in the study group. The patients’ mean age was 38.17 ± 11.86 years old (range 16–73 years). The study included 78 patients with migraine without aura and 22 patients with migraine with aura. Among patients, 70 suffered episodic migraine (EM) and 30 had chronic migraine (CM) (93% females; mean age 38.8 years). The EM group included 61 female (87%) and nine male patients with a mean age of 37.9 ± 11.07, and the CM group had 28 female and two male patients with mean a age 38.8 ± 13.72. Medication overuse headaches were, in general, rare, but more common among chronic migraine (3) than episodic migraine (3) patients (10% vs. 4%). The migraine group demographic details are presented in Table 1.

The migraineurs were asked to complete the EQ-5D-5L questionnaire according to the Bioethical Committee Agreement. The study was accepted by the Bioethical Committee of the Medical University of Warsaw (AKBE/100/2022). Oral informed consent was obtained from all the individual participants included in the study.

#### 2.1.2. Control Group

The control group (n = 100), matched for sex and age group, was created based on the results of the population norms study for the EQ-5D-5L in the general population of Poland (N = 3963) [34,35,36]. Each patient was assigned a virtual respondent from the general population matched in terms of sex and age group (seven age groups were distinguished: 18–24, 25–34, 35–44, 45–54, 55–64, 65–74, and 75+ years). The results of the quality of life of virtual patients from the control group were the mean values for a given sex and a given age group published in [34] (distribution of limitations within the EQ-5D-5L dimensions), [35] (mean EQ-5D-5L Index value) and [36] (the mean EQ VAS score).

### 2.2. Instrument

General HRQoL was assessed using the five-level version of the EQ-5D questionnaire [31]. The EQ-5D-5L consists of two parts:the descriptive system;the visual analogue scale.

The descriptive system includes five dimensions:mobility (MO);self-care (SC);usual activities (UA);pain/discomfort (PD);anxiety/depression (AD).

Each dimension has five potential levels: from ‘1’ (no limitations) to ‘5’ (extreme limitations). The EQ-5D-5L descriptive system defines 3125 potential different health states. Each health state can be assigned a single number representing the overall health score—the EQ-5D-5L Index. In the present study, to estimate the EQ-5D-5L Index values, the Polish EQ-5D-5L Index value set obtained with direct measurement methods (time trade-off, discrete choice experiment) in the Polish population was used [30]. The EQ-5D-5L Polish index values range from −0.590 to 1.0 (full health). The EQ-5D visual analogue scale (EQ VAS) is a vertical scale numbered from 0 (the worst health you can imagine) to 100 (the best health you can imagine) and is used for the subjective assessment of HRQoL.

We did not use any disease-specific instrument.

### 2.3. Statistical Methods

For continuous variables (EQ-5D-5L Index, EQ VAS scores), the mean values with standard deviation and 95% confidence intervals were estimated. We used Wilcoxon’s signed-ranks test to determine the differences between the analyzed groups [37]. Multiple linear regression was used to examine the associations of demographic characteristics (sex, age group) and diagnosis of migraine (episodic, chronic, no migraine) with the EQ-5D-5L Index and EQ VAS scores. All variables were entered into the models as categorical variables. Regression coefficients were presented together with information about the level of statistical significance. The analysis was conducted using StatsDirect 3.3.5 (StatsDirect Ltd., Altrincham, UK) statistical software.

## 3. Results

### 3.1. Migraine Patients versus the General Population

Patients with migraine had worse HRQoL than the matched general population control group for all three primary endpoints of the EQ-5D-5L questionnaire: dimensions, EQ-5D-5L Index and EQ VAS.

Figure 1 compares the limitations within the five EQ-5D-5L dimensions between migraine patients and the matched general population cohort. Level 1 means no limitations on a given domain, and level 5 means extreme limitations. According to the previously published data [34], in the general population of Poland, the dimension with the lowest number of limitations is self-care, and the dimensions with the highest number of limitations are pain/discomfort and anxiety/depression. The visual analysis of the graphs indicated that within the two dimensions (mobility, self-care), there are no significant differences in the rate of limitations between migraine patients and the general population. Migraine diagnosis has the most significant impact on the increase in the incidence of limitations within the pain/discomfort dimension (the largest area between the two curves), followed by the anxiety/depression and usual activities dimensions.

Patients with migraine, compared to the general population, are characterized by a worse HRQoL in terms of both the EQ-5D-5L Index—mean difference of 0.062 (SD 0.150; *p* < 0.001) and the EQ VAS—mean difference of 7.9 (SD 18.5; *p* < 0.001) points (Table 2; Figure 2A and Figure 3A).

### 3.2. Episodic and Chronic Migraine Patients versus the General Population

Patients with episodic migraine, compared to the general population, had an average of 4.7 (SD 16.5) points lower on the quality-of-life assessment using EQ VAS (*p* < 0.05; Figure 3B). The difference in the EQ-5D-5L Index (0.034, SD 0.105) was not significant but indicated a statistical tendency (*p* = 0.06; Table 2; Figure 2B).

Multiple linear regression showed that the diagnosis of episodic migraine, regardless of the influence of the patient’s sex and age, resulted in a significant mean reduction in the EQ-5D-5L Index by 0.033 and the EQ VAS scores by 4.65 points (Table 3).

Patients with chronic migraine, compared to the general population, had a mean 0.130 (SD 0.211) lower EQ-5D-5L Index score (*p* < 0.001; Table 2; Figure 2C) and a mean of 15.4 (SD 20.7) points lower on the EQ VAS score (*p* < 0.001; Table 2; Figure 3C).

Multiple linear regression indicated that the diagnosis of chronic migraine, regardless of the influence of the patient’s sex and age, resulted in a significant mean reduction in the EQ-5D-5L Index by 0.132 and the EQ VAS by 15.57 points (Table 3).

## 4. Discussion

Our study of migraineurs’ general health-related quality of life based on the EQ-5D-5L responses, together with the comparison with the representative sample of Polish citizens, shows the devastating impact of the disease on migraine patients, with a particular indication of chronic migraine. The impact of the disease reducing the quality of life is mainly manifested in the feeling of discomfort, pain, anxiety and a significantly depressed mood. This affects the functioning of patients in their personal, family, social and professional activities. While mobility and self-service in migraine patients are not impaired, the other dimensions make migraine one of the most disabling lifelong diseases. It is true that those most affected by disability are those of the most active and productive age. Similar results have been published, but the authors use different tools. Ahadi et al. [38] measured the quality of life in migraineurs mainly based on the MSQ, which is a condition-specific measure that also uses a generic preference-based measure (EQ-5D-5L), and afterwards maps an algorithm to estimate health-state utility values. The proposed MSQ mapping algorithm would be suitable for estimating health state utilities in trials of patients with migraine that contain MSQ scores but lack utility values. There are increasing demands toward the employment of cost-utility in healthcare decisions regarding resource allocation and decision-making, where effectiveness is measured in the quality-adjusted life years (QALY). Estimation of QALY requires measurement of health-related utility. In the study of Ahadi et al. [38] the same prediction problem in all migraineurs, episodic and chronic migraine were proved. They concluded that the preferred MSQ mapping algorithm would be suitable for estimating health-state utilities in trials of patients with migraine that contain MSQ scores but lack utility values. We confirmed the problem of worse HRQoL in all migraine patients, which was much more pronounced in patients with chronic migraine (for all three primary endpoints of the EQ-5D-5L questionnaire: dimensions, EQ-5D-5L Index and EQ VAS) vs. episodic migraine patients (EQ-5D-5L Index was not significant but indicated a statistical tendency). On the other hand, Rupel et al. [39] found in their systemic review of the central and eastern Europe published literature that the EQ-5D is either completely lacking or has very scarce data in some neurological areas with a significant social burden, such as a migraine. Thirty-six articles describing the results of 38 samples of patients and a total of 13,005 patients, as well as most studies from Hungary were included in this review. EQ-5D utility scores were reported in more than 90% of articles. With multiple sclerosis being the most represented disease, the average utility scores ranged from 0.49 in Austria to 0.80 in Poland, with a weighted average of 0.69. The EQ VAS scores for MS ranged from 39.0 in the Czech Republic to 72.0 in Poland, with a weighted average of 59.1. Multiple sclerosis, epilepsy and essential tremor patients estimated their HRQoL among the highest. Jankowska et al. recently developed health-related quality-of-life norms for patients with self-reported diabetes, based on a large representative sample of the general Polish population, using the EQ-5D-5L [40], proving that diabetes leads to HRQoL deterioration. In the diabetic population, the most significant HRQoL reduction is experienced by older patients with a basic level of education. That is, using an appropriate patient quality-of-life questionnaire helps to improve patients care.

So far, the EQ-5D-5L has not often been used in patients with migraine. The current study is the first of this kind in Poland. According to our results, the EQ-5D-5L confirms that in migraine, HRQoL is significantly impaired and may indicate therapeutic directions for improvement. The results of our study may be another reliable and relevant source of data on the burden of migraine. There are increasing demands for studies on the cost-effectiveness of allocating resources for disease prevention and treatment strategies. Some studies have assessed the quality of life in migraineurs based on condition-specific measures of quality of life. However, it is important to understand outcomes in terms of their effects on overall health status, to examine the relative value of various treatments available for migraine. There are not many studies worldwide that have mapped migraine-condition-specific questionnaires to utility. The use of generic questionnaires, such as the EQ-5D, and especially EQ-5D-5L, enables the comparison of patient groups with the general population of the country or between EM vs. CM, and the objective assessment of the burden of the disease.

Our study has some limitations. The study group is not large and comes from one center. On the other hand, one center and one physician qualifying all patients according to the diagnostic criteria and inclusion criteria assures the consistency and reliability of the results. Using only one questionnaire—the EQ-5D-5L—in the study may be a limitation of the study.

## 5. Conclusions

To sum up, migraine is a disease that disrupts daily functioning, and as a lifelong disease, plays a role in every aspect of it. Proving its negative impact on many aspects helps to make decisions about treatment, especially in the context of the design and reimbursement of drugs.

## Figures and Tables

**Figure 1 jcm-11-06925-f001:**
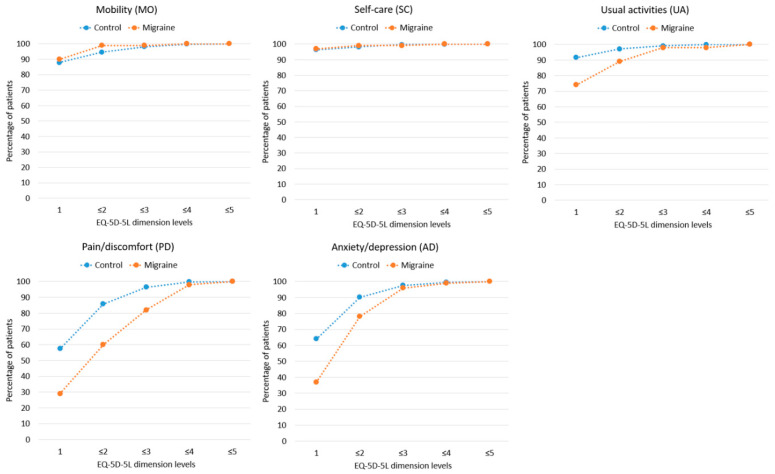
EQ-5D-5L dimensions in patients with migraine (N = 100) compared to control from the general population (N = 100; dashed lines are means of graphical visualization).

**Figure 2 jcm-11-06925-f002:**
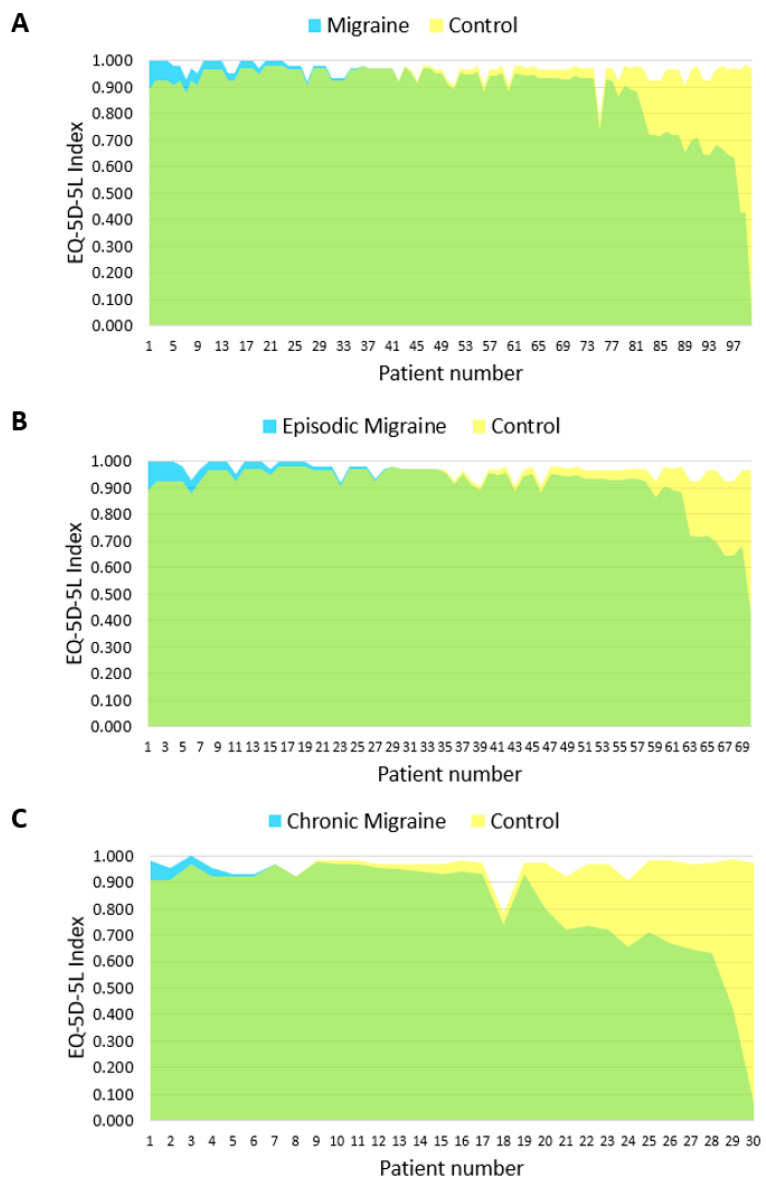
The burden of (**A**) migraine, (**B**) episodic migraine and (**C**) chronic migraine according to the EQ-5D-5L Index. Yellow color denotes health utility lost due to the disease in comparison to controls from the general population.

**Figure 3 jcm-11-06925-f003:**
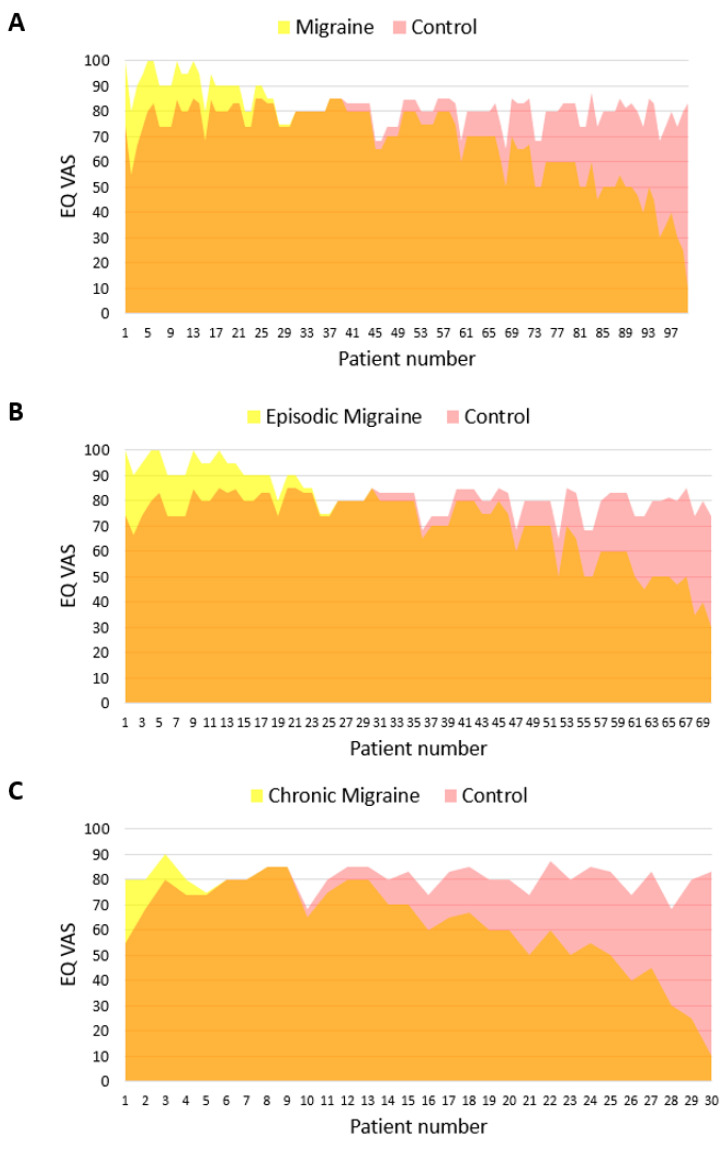
The burden of (**A**) migraine, (**B**) episodic migraine and (**C**) chronic migraine according to the EQ VAS. Pink color denotes the quality of life lost due to the disease in comparison to controls from the general population.

**Table 1 jcm-11-06925-t001:** Characteristics of the studied population.

		All Migraine Patients	Episodic Migraine	Chronic Migraine
N		100	70	30
Age, years	mean (SD)	38.2 (11.9)	37.9 (11.1)	38.8 (13.7)
	Range	16–73	16–65	17–73
Sex, females, n (%)		89 (89)	61 (87)	28 (93)
Type of migraine	without aura (%)	78 (78)	51 (72)	27 (90)
	with aura (%)	22 (22)	19 (28)	3 (10)
Medication overuse headache (MOH), n (%)		6 (6)	3 (4)	3 (10)

**Table 2 jcm-11-06925-t002:** EQ-5D-5L outcomes in patients with migraine and control group.

	Migraine			Episodic Migraine			Chronic Migraine		
	Patients	Control	Difference (Control—Patients)	Patients	Control	Difference (Control—Patients)	Patients	Control	Difference (Control—Patients)
**N**	100	100		70	70		30	30	
**EQ-5D-5L Index**									
mean (SD)	0.892 (0.149)	0.955 (0.031)	0.062 (0.150) *	0.922 (0.107)	0.956 (0.026)	0.034 (0.105) **	0.822 (0.204)	0.952 (0.041)	0.130 (0.211) *
95%CI	0.862–0.922	0.948–0.961	0.033–0.092	0.896–0.947	0.949–0.962	0.009–0.059	0.746–0.898	0.936–0.967	0.051—0.209
**EQ VAS**									
mean (SD)	71.1 (18.7)	79.1 (5.9)	7.9 (18.5) *	74.5 (17.5)	79.2 (5.3)	4.7 (16.5) ***	63.4 (19.4)	78.8 (7.1)	15.4 (20.7) *
95%CI	67.4–74.8	77.9–80.2	4.3–11.6	70.3–78.6	77.9–80.5	0.8–8.7	56.2–70.6	76.1–81.4	7.6–23.1
**Severity Index**									
mean (SD)	7.8 (2.5)	6.5 (0.9)	−1.3 (2.6) *	7.3 (2.0)	6.5 (0.7)	−0.9 (1.9) *	8.9 (3.3)	6.6 (1.1)	−2.3 (3.6) *
95%CI	7.3–8.3	6.3–6.7	−1.8–−0.8	6.8–7.8	6.3–6.6	−1.3–−0.4	7.7–10.1	6.2–7.0	−3.7–−1.0

* *p* < 0.001; ** *p* = 0.06; *** *p* < 0.05.

**Table 3 jcm-11-06925-t003:** Influence of demographic factors and migraine status on HRQoL outcomes.

	N (%)	EQ-5D-5L Index			EQ VAS		
		Mean (S.D.)	Multiple Linear Regression Coefficients	*p*-Value	Mean (S.D.)	Multiple Linear Regression Coefficients	*p*-Value
**Intercept**	200 (100)	0.923 (0.112)	0.873	*p* < 0.0001	75.10 (14.37)	67.50	*p* < 0.0001
**Sex**							
Male	22 (11)	0.920 (0.130)	-	-	78.06 (14.60)	-	-
Female	178 (89)	0.924 (0.110)	−0.001	*p* = 0.964	74.74 (14.34)	−5.04	*p* = 0.117
**Age group**							
18–24 years	28 (14)	0.929 (0.125)	0.098	*p* = 0.104	81.13 (10.95)	23.36	*p* = 0.002
25–34 years	52 (26)	0.943 (0.136)	0.097	*p* = 0.095	79.01 (14.39)	18.95	*p* = 0.008
35–44 years	60 (30)	0.928 (0.105)	0.087	*p* = 0.143	75.19 (13.56)	16.51	*p* = 0.023
45–54 years	40 (20)	0.905 (0.087)	0.061	*p* = 0.305	71.12 (14.75)	12.09	*p* = 0.098
55–64 years	16 (8)	0.901 (0.073)	0.064	*p* = 0.304	64.66 (13.23)	5.93	*p* = 0.434
65+ years	4 (2)	0.832 (0.088)	-	-	62.45 (13.22)	-	-
**Migraine diagnosis**							
No migraine	100 (50)	0.955 (0.031)	-	-	79.06 (5.87)	-	-
Episodic migraine	70 (35)	0.922 (0.107)	−0.033	*p* = 0.042	74.46 (17.47)	−4.65	*p* = 0.019
Chronic migraine	30 (15)	0.822 (0.204)	−0.132	*p* < 0.0001	63.40 (19.41)	−15.57	*p* < 0.0001

## Data Availability

The data can be found in the documentation collected in the Headache Outpatient Clinic.

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
