# Peer review of "Health-Related Quality of Life in Migraine: EQ-5D-5L-Based Study in Routine Clinical Practice"

_jcm, 2022, doi:10.3390/jcm11236925_

Round 1

Reviewer 1 Report

1. Could not find decoding of abbreviation for:  mMIDAS-P

2. In the 2.1. Study groups:  "The cross-sectional study was conducted between September 2019 and December 20220 ...". In turn, in 2.1.1. Study group with migraine:  "The study was accepted by the Bioethical Committee ... (AKBE/100/2022)".                                                            Was the study accepted by the Bioethical Committee after the study was carried out? 

3. In the References list:                                                                    Reference No 38 "The impact of symptoms on quality of life before and after diagnosis of coeliac disease: the results from a Polish population..." seems not to be relevant to the current research.

4. In 2.1.1. Study group with migraine: "Oral informed concent was obtained from all individual participants included in the study".                 Oral informed concent could be enough, if personal data of the participants were not obtained. If personal data was collected, this data should be protected and written informed concent may be needed.

Author Response

Dear Reviewer

Thank you very much for your thorough evaluation of the work and valuable comments. In response, we explain:

  1. Could not find decoding of abbreviation for:  mMIDAS-P

Answer: we have expanded the acronym mMIDAS-P as “modified Migraine Disability Assessment in Portuguese population” (line 71 and line 260)

  1. In the 2.1. Study groups:  "The cross-sectional study was conducted between September 2019 and December 20220 ...". In turn, in 2.1.1. Study group with migraine:  "The study was accepted by the Bioethical Committee ... (AKBE/100/2022)".  Was the study accepted by the Bioethical Committee after the study was carried out? 

Answer: Yes - The study has been approved by the Bioethics Committee as, a non-invasive and use of tests carried out during the routine (retrospective) visits of patients for scientific and publication purposes.

  1. In the References list:  Reference No 38 "The impact of symptoms on quality of life before and after diagnosis of coeliac disease: the results from a Polish population..." seems not to be relevant to the current research.

Answer: We have deleted this sentence and reference no 38 and changed the numbering of the remaining references.

  1. In 2.1.1. Study group with migraine: "Oral informed concent was obtained from all individual participants included in the study".  Oral informed concent could be enough, if personal data of the participants were not obtained. If personal data was collected, this data should be protected and written informed concent may be needed.

Answer: According to the Bioethics Commission's decision, oral consent was sufficient - personal date of the participants were not obtained.

Reviewer 2 Report

I had the pleasure to review the articel „Health-related quality of life in migraine: EQ-5D-5L-based study in routine clinical practice“ written by Izabela Domitrz and Dominik Golicki. The topic of the paper is interesting, it is well written and the conclusions drawn are supported by the performed analysis.

While reading and working through the manuscript, I noticed the following aspects:

1) Throughout the manuscript, the EQ-5D-5L questionnaire is described as a tool to assess health related quality of Life (HRQoL). In fact, the EQ-5D only measures general QOL, as it is not a disease-specific measurement tool. Comparison with a collective of healthy patients can ultimately lead to the inference of HRQoL, but this should be separated more in the manuscript.

2) Was there a specific reason not to use a migraine-specific questionnaire, such as the MSQ (Migraine Specific Quality of Life Questionnaire)? If so, the reason should be stated. Moreover, the use of a non-specific QoL questionnaire should be noted as limitation. 

3) This is more a suggestion than a criticism of the manuscript itself: In Patients with epilepsy, comparing the EQ-5D and a disease specific questionnaire (QOLIE31) relevant disease aspects could be identified that were not reflected in the EQ-5D values, e.g. seizure frequency as important clinical Aspect of epilepsy (https://doi.org/10.1016/j.yebeh.2022.108554). Are the available data sufficient to perform a similar analysis? (e.g. correlation between frequency of migraine attacks and EQ-5D).

Minor aspects:

4) Figure 1: From a statistical point of view, the connecting lines between the measured points are not valid, as they suggest a linear relationship that has not been proven. I would recommend to remove the lines. Alternatively, they could be shown as dashed lines and explained in the legend as a means of graphical visualization.

5) Figure 1-3: The images are generally in a very poor resolution and look blurred.

6) Section 2.1.2 and Table 3: The term "gender" refers to gender identification. Since I assume that the biological sex is indicated here, I would use the term "sex" consistently, as in the tables.

7) A multivariate regression analysis based on the data is surely possible, but not necessarily meaningful, especially due to the low number of variables examined and their different data levels (nominal: gender, migraine diagnosis vs: ratio: age, here distributed in several age groups). Alternatively, with regard to age, a correlation with subsequent Z-transformation could be more meaningful, and with regard to sex (not gender -> Table 3, see comment 6) as well as migraine diagnosis, an adequate statistical test with subsequent post-hoc correction. However, this is only a suggestion.

8) Since routinely collected clinical data were used in the analysis, the RECORD guidelines (REporting of studies Conducted using Observational Routinely-collected Data) should be applied and addressed appropriately to support the quality of the data and the analysis.

Author Response

Dear Reviewer

Thank you very much for your thorough evaluation of the work and valuable comments. In response, we explain:

  • Throughout the manuscript, the EQ-5D-5L questionnaire is described as a tool to assess health related quality of Life (HRQoL). In fact, the EQ-5D only measures general QOL, as it is not a disease-specific measurement tool. Comparison with a collective of healthy patients can ultimately lead to the inference of HRQoL, but this should be separated more in the manuscript.

Answer: Thank you for this valuable comment. We of course agree that EQ-5D is a generic (not disease specific instrument) and it measures general HRQoL. Still, we believe it measures HRQoL, not QoL, as QoL includes such areas as spirituality, happiness, life perception, satisfaction, economic safety, and so on.

Following your suggestion we modified some paragraphs. We added to paragraph 2.2 underlining that we measured ‘General HRQoL’. We also added the statement that we ‘We did not use any disease specific instrument’. And a statement in the Discussion section that ‘the study was on the general health-related quality of life’.

  • Was there a specific reason not to use a migraine-specific questionnaire, such as the MSQ (Migraine Specific Quality of Life Questionnaire)? If so, the reason should be stated. Moreover, the use of a non-specific QoL questionnaire should be noted as limitation. 

Answer: The aim of the study was to use a new questionnaire EQ-5D-5L in Poland, which had not been used so far to assess patients with migraine, therefore the focus was on EQ-5D-5L, not MSQ.

We have added in discussion: “Using only one questionnaire – EQ-5D-5L - in the study may be a limitation of the study.” (line 245)

  • This is more a suggestion than a criticism of the manuscript itself: In Patients with epilepsy, comparing the EQ-5D and a disease specific questionnaire (QOLIE31) relevant disease aspects could be identified that were not reflected in the EQ-5D values, e.g. seizure frequency as important clinical Aspect of epilepsy (https://doi.org/10.1016/j.yebeh.2022.108554). Are the available data sufficient to perform a similar analysis? (e.g. correlation between frequency of migraine attacks and EQ-5D).

Answer: The study compared EQ-5D-5L in a group of patients with chronic migraine (in which the frequency of headaches and migraine attacks is higher than in episodic migraine) with EQ-5D-5L of episodic migraine. The analysis of the frequency and severity of attacks in episodic migraine and EQ-5D-5L results is another element of the study that we plan to publish in the next step.

  • Figure 1: From a statistical point of view, the connecting lines between the measured points are not valid, as they suggest a linear relationship that has not been proven. I would recommend to remove the lines. Alternatively, they could be shown as dashed lines and explained in the legend as a means of graphical visualization.

Answer: Thank you for this valuable comment. Following your advice we changed solid lines to dashed lines. We also added comment to the Figure title (“dashed lines are means of graphical visualization”).

  • Figure 1-3: The images are generally in a very poor resolution and look blurred.

Answer: Following your comment, we have improved the quality of Figures 1-3.

  • Section 2.1.2 and Table 3: The term "gender" refers to gender identification. Since I assume that the biological sex is indicated here, I would use the term "sex" consistently, as in the tables.

Answer: We have changed “gender” into “sex” in: table 3, sections 2.1.2 and 3.2

  • A multivariate regression analysis based on the data is surely possible, but not necessarily meaningful, especially due to the low number of variables examined and their different data levels (nominal: gender, migraine diagnosis vs: ratio: age, here distributed in several age groups). Alternatively, with regard to age, a correlation with subsequent Z-transformation could be more meaningful, and with regard to sex (not gender -> Table 3, see comment 6) as well as migraine diagnosis, an adequate statistical test with subsequent post-hoc correction. However, this is only a suggestion.

Answer: Thank you for your valuable comments. We have discussed your suggestions in details. Finally, we have decided we prefer to stay with a little bit simpler analysis showed originally. Nevertheless, we appreciate your in-depth view and original ides. 

  • Since routinely collected clinical data were used in the analysis, the RECORD guidelines (REporting of studies Conducted using Observational Routinely-collected Data) should be applied and addressed appropriately to support the quality of the data and the analysis.

Answer: We have taken into account the RECORD guidelines when creating the manuscript.
